# Evaluation of Microscopic Damage of PEEK Polymers under Cyclic Loadings Using Molecular Dynamics Simulations

**DOI:** 10.3390/polym14224955

**Published:** 2022-11-16

**Authors:** Shun Iwamoto, Yutaka Oya, Jun Koyanagi

**Affiliations:** 1Department of Materials Science and Technology, Graduate School of Tokyo University of Science, 6-3-1 Niijuku, Katsushika-ku, Tokyo 125-8585, Japan; 2Research Institute for Science & Technology, Tokyo University of Science, 6-3-1 Niijuku, Katsushika-ku, Tokyo 125-8585, Japan; 3Department of Materials Science and Technology, Tokyo University of Science, 6-3-1 Niijuku, Katsushika-ku, Tokyo 125-8585, Japan

**Keywords:** molecular dynamics, thermoplastic resin, cyclic loading, entropy, void

## Abstract

Full-atomic molecular dynamics simulations were conducted to investigate the time evolution of microscopic damage in polyetheretherketone (PEEK) polymers under cyclic loading conditions. Three characteristics were used to quantify microscopic damage: entropy, distribution of the end-to-end distance of polymers, and the volume fraction of voids. Our results show that the degree of disentanglement of polymers and the volume fraction of voids increase with cyclic loading, which may lead to entropy generation. Uniaxial tensile strength simulations of the polymer system before and after cyclic loading were performed. The tensile strength after cyclic loading was lower than that before loading. Furthermore, two systems with the same entropy and different loading histories showed almost the same strength. These results imply that entropy generation is expressed as the total microscopic damage and can potentially be employed for effective evaluation of the degradation of material characteristics.

## 1. Introduction

Thermoplastic polymers are being increasingly used as the matrix materials of composites instead of thermosetting polymers. (In this paper, composites based on thermoplastic and thermosetting polymers are referred to as CFRTPs and CFRPs, respectively.) This is because CFRTPs have been used in the automotive and infrastructure industries due to excellent impact resistance, moldability, and recyclability [1,2]. Understanding the mechanism of resin damage is critical for improving resin durability for future applications.

Because of their complex fracture behavior, it has been difficult to quantitatively predict the durability and residual life of thermoplastic polymers [1,3]. Thermoplastic resins used in aircraft and automobiles are subjected to mechanical loading, especially cyclic loading, which causes microscopic damage (cavities called voids) accumulation inside the resin, leading to sudden macroscopic failure [4,5]. The fracture mechanism of thermoplastic resins has not yet been revealed because it is difficult to experimentally measure microscopic damage.

To date, the fatigue and fracture behaviors of CFRTPs have been extensively investigated [6,7,8,9,10,11]. Koyanagi et al. reported that the mechanical characteristics of the matrix have a significant impact on those of the composite. Therefore, it is important to understand the mechanisms of microscopic damage and fracture behavior in thermoplastic resins to predict the durability and residual life of CFRTPs [12].

A molecular dynamics (MD) simulation is an effective approach for reproducing and evaluating the microscopic damage of thermoplastic resin. This is because MD simulations allow us to obtain the morphology of polymers and the thermodynamic properties based on each atomic motion, which means that MD simulations have much higher spatial and temporal resolutions than those of standard experiments with respect to mechanical properties. In previous research, polymer/carbon fiber interfaces have been investigated to improve the mechanical properties of CFRPs using MD simulation [13,14,15,16]. Various elongations and cyclic loadings were conducted to study the mechanical properties of CFRPs by MD simulation [17,18,19,20,21,22]. These studies demonstrated that MD simulation is indispensable for predicting the material characteristics and revealing the atomistic-scale mechanisms that cause these properties. In this study, we apply MD simulations to evaluate the microscopic damage of polymers under cyclic loading, which is important for the long-term durability of structural materials in transport. Based on the following previous studies, this study focuses on the generation of entropy for the quantitative evaluation of damage.

The entropy-based destruction rule has been proposed to represent material damage. In this rule, material failure is defined by the critical entropy inherent in a material [4,23,24,25]. The entropy is calculated by dividing the dissipation energy by the absolute temperature, where the dissipation energy is calculated from the stress–strain diagram [26]. The entropy-based destruction rule has traditionally been used in metallic materials such as aluminum and iron [27,28,29,30]. Recently, this rule has also been applied to polymer materials under tensile and cyclic loading. However, most numerical studies applying this rule are based on continuum models such as the finite element method (FEM), which considers phenomena only over the millimeter scale [12]. For microscopic damage, Takase et al. evaluated the entropy generation of polyamide 6 in a tensile test using MD simulations [4]. This paper reports that polymers with different loading histories have nearly identical entropies at fracture, which suggests that it is possible to reproduce resin damage through MD and to evaluate microscopic damage using entropy. This study investigates the relationships among entropy, voids, and the degree of disentanglement.

The main factors underlying entropy generation are thought to be the disentanglement of polymers and the formation of voids, as investigated in previous studies [31,32]. Voids are often regarded as material damage because they can become the starting points of material failure [5]. Takase et al. found that both the void volume and entropy increase during the deformation of the polymer material [4]. Thus, the evaluation of the number of voids and the void volume fraction on a microscopic scale can lead to the estimation of damage in polymer materials. The end-to-end distance has also been employed to discuss the mechanical properties of resin under loading conditions [17,18,19,20]. This is because the degree of disentanglement, which degrades the mechanical properties of polymers, can be quantitatively evaluated using the end-to-end distance. Therefore, the end-to-end distance distribution of the polymer under loading conditions is important for evaluating polymer damage.

After the cyclic loading tests, we performed a uniaxial tensile simulation to evaluate the residual strength, which is an important indicator for determining the remaining life of a material [23,26]. Sato et al. attempted to predict the residual strength of CFRPs using the entropy-based destruction rule [23]. By determining the residual strength of the resin, it would become possible to verify whether the resin is actually damaged, that is, whether the entropy increase reflects the resin damage in the MD simulation.

The purpose of this study is to reproduce the cyclic loading test for thermoplastic polymers and to investigate microscopic damage using entropy, voids, and end-to-end distance in MD simulations. The remainder of this paper is organized as follows. The next section is devoted to the simulation methods for obtaining the equilibrium structure, reproducing cyclic loading, and evaluating microscopic damage to polymers. In the third section, the simulation results and discussion about the material characteristics associated with microscopic damage, that is, the entropy, end-to-end distance, volume fraction of the void, and residual strength, are presented. We conclude the paper in the final section.

## 2. Method

### 2.1. Modeling and Relaxation Process

We conducted cyclic loading tests using full-atomic molecular dynamics simulations for the quantitative evaluation of microscopic damage to PEEK resin. To determine the molecular structure of PEEK for the MD simulation, Marvin Sketch and PolyPerGen were successively used [33,34]. Figure 1 and Figure 2 show the chemical structure of polyetheretherketone (PEEK) and the corresponding molecular structure after structural optimization calculations, respectively. The degree of polymerization was set to 10, and the number of atoms in the system was approximately 50,000.

Before the cyclic loading tests, the system was relaxed to achieve the equilibrium state at room temperature (300 K) and atmospheric pressure (1 atm) in four steps. First, all molecular chains were randomly arranged in the simulation cell. Second, relaxation calculations were conducted under the *NPT* ensemble at *p* = 1 atm and *T* = 650 K, where *p* is the pressure and *T* is the temperature of the system. Third, the system was annealed at *T* = 300 K (room temperature), *p* = 1 atm, and a cooling rate of 70 K/ns. Finally, the equilibrated system was obtained by relaxation calculations for 2 ns under the *NPT* ensemble at *T* = 300 K and *p* = 1 atm. Figure 3 shows a snapshot of the equilibrium system. Based on the equilibrium calculation, the density of the system was 1.19 g/cm^3^, which is an appropriate value corresponding to that obtained by experiment (1.3 g/m^3^) [35,36]. Furthermore, time evolutions of the total potential energy and the volume fraction of voids confirmed that the system had fully reached equilibrium.

All calculation processes in the MD simulation were conducted using the GROMACS software [37], and an all-atom optimized potential for liquid simulation (OPLS-AA) force field was employed [38]. For the electrostatic potential charge, density functional calculations were conducted using B3LYP/6-31G (Hamiltonian/basis set) [39,40].

### 2.2. Cyclic Loading

Cyclic loading simulations were performed for two different strain rates (±5.0 × 10^8^/s and ±5.0 × 10^9^/s). The system was deformed in the z-direction while keeping the x and y lengths constant. The minimum and maximum values of strain in the z-direction were set to 0.1 and 0.0, respectively. The total simulation time was fixed at 40 ns, corresponding to 100 cycles at a strain rate of 5.0 × 10^8^/s and 1000 cycles at 5.0 × 10^9^/s. The microscopic damage of PEEK resin was quantitatively evaluated in terms of four material characteristics: entropy generation, total amount and distribution of voids, end-to-end distance of the polymer chain, and residual strength. Figure 4 shows the relationship between time and strain in the z-direction during each cyclic loading for 5.0 × 10^8^/s (a) and 5.0 × 10^9^/s (b).

### 2.3. Evaluation of Resin Damage

Previous research has suggested that the entropy of material fracture does not depend on the loading conditions [41]. Therefore, entropy generation represents the degree of fracture. To estimate the entropy generation under cyclic loading, we used the “mechanical method,” where the entropy is calculated using the following equation:(1)$$E_{mechanical}=\overset{t}{\underset{0}{\int }}\frac{W_{d}}{T}dt$$
where $E_{mechanical}$, t, $W_{d}$, and T are the entropy generation [J/Km^3^], time for cyclic loading [s], dissipation strain energy [Pa], and absolute temperature [T], respectively. In our simulation, $W_{d}$ was quantitatively measured using the stress–strain relationship, as reported in previous work [4]. Equation (1) describes the entropy generation in the quasi-static process, which is much slower than the time scale used in standard MD simulations. To confirm the quantitative correctness, we have previously compared the entropy generation obtained by this approach to that obtained by energy balance for various tensile simulations. In the results, the entropies obtained by these two different approaches were quantitatively in agreement [4]. Furthermore, we also confirmed that this approach can estimate the entropy on the same order of magnitude as experimental results by different scanning calorimetry [24]. Therefore, the approach used in this study is confirmed to be reliable enough to estimate the entropy generations of polymers, at least for the tensile deformation.

The conformation entropy of a single ideal polymer chain is closely related to its end-to-end distance, R, defined by
(2)$$R=\left|\overset{n}{\underset{i=1}{\sum }}b_{i}\right|$$
where $b_{i}$ is the bond vector between the (i−1)-th and i-th atoms, and i=0 and i=n denote two atoms at the endpoints of the main polymer chain. Basically, R is inversely related to the conformational entropy of polymers. The R of the isolated polymer chain statistically takes a Gaussian distribution with a peak position of R=0, and the free energy contributed by the conformational entropy is proportional to $R^{2}$ [42]. This distribution changes because of the mutual entanglements among the polymer chains. Therefore, the degree of entanglement was quantitatively evaluated using the distribution of R.

The number of voids and the void volume fraction were calculated using the Open Visualization Tool (OVITO) and ParaView [43,44]. The alpha shape method was employed to calculate the void volume fraction [45]. The number of voids and the void volume fraction were measured at strain 0 at the same volume points. The measurements were conducted every 10 cycles at low frequency (strain rate 5.0 × 10^8^/s) and every 100 cycles at high frequency (strain rate 5.0 × 10^9^/s). In this study, we counted only voids with a volume larger than 60 ${Å}^{3}$, which is the smallest volume that can be measured by the experiment with the positron annihilation method. It should be noted that all systems under the same strain had the same volume. Considering all voids smaller than 60 ${Å}^{3}$, the total void volume was the same for all systems at the same strain.

## 3. Results and Discussion

### 3.1. Stress–Strain Curves

We conducted cyclic loadings at different strain rates to investigate the dependence of the resin damage on the strain rate. Figure 5 shows the stress–strain curves for strain rates of (a) 5.0 × 10^8^/s and (b) 5.0 × 10^9^/s. We observed that both stress–strain curves (Figure 5a,b) shift to the bottom right, and inelastic strain increases with an increase in the number of cycles. This indicates that the resin was damaged by the cyclic loading. By comparing Figure 5a,b, we observed that the stress–strain curves for the 10th and 100th cycles at a strain rate of 5.0 × 10^8^/s were similar to those for the 100th and 1000th cycles at a strain rate of 5.0 × 10^9^/s. This result suggests that the amount of resin damage is determined by the simulation time and not the number of cycles. This may be because the strain rates used in the MD calculations were significantly larger than those used in standard tensile experiments.

### 3.2. Entropy Generation

We obtained the dissipation energy from the stress–strain curves. The entropy generation, $E_{mechanical}$, was calculated using Equation (1). The dissipation energy, $W_{d}$, in Equation (1) was equal to the total area enclosed by the stress–strain curve, owing to the stress loading and unloading. Figure 6 shows the time development of entropy generation under cyclic loading with different strain rates. The two curves coincide with each other, indicating that the simulation time is more significant in determining the entropy generation than the number of cycles, as in the case of the stress–strain curves. It can be observed that entropy increases rapidly in the first 4000 ps, corresponding to the first 10 cycles at a strain rate of 5.0 × 10^8^/s and 100 cycles at 5.0 × 10^9^/s. Thereafter, the rate of entropy generation decreases with time. These results suggest that microscopic damage increases rapidly in the first 4000 ps and then slows down. To confirm this damage expansion, the end-to-end distance of each polymer and void was analyzed as follows.

### 3.3. End-to-End Distance Distribution

The end-to-end distance of each polymer under cyclic loading was calculated using Equation (2) to investigate the degree of entanglement. An increase in R corresponds to an increase in entanglement, and a decrease in R corresponds to a decrease in entanglement. Statistically, R decreases if the polymers are disentangled. Figure 7a shows the frequency distribution of R for a strain rate of 5.0 × 10^8^/s. As the cyclic loadings increase, some peaks shift from the higher R side to the lower side and their frequency decreases, indicating the disentanglement of polymers. As is well known statistically, for an ideal polymer chain, the conformation entropy and the end-to-end distance are closely related to each other; the entropy increases as R decreases. Therefore, our results suggest that one of the important factors in entropy generation shown in Figure 6 is the increase in the conformation entropy of polymers due to disentanglements.

### 3.4. Void Analysis

The time evolutions of the number and volume fraction of voids are shown in Figure 8a,b, respectively. For both of the cyclic loading cases, the number of voids is almost constant with respect to time. However, the volume fraction of voids increases monotonically with time. These results indicate that each void grows with time in a nucleation-like manner, in which newly formed smaller voids are absorbed by larger ones. Figure 8b also shows that the void volume at a strain rate of 5.0 × 10^8^/s rapidly increases in the first cycles and then increases slowly. This trend is qualitatively consistent with the time development of entropy generation, which suggests that the degree of microscopic failure is closely related to the volume fraction of the void. Note that the simple relaxation case, seen in Figure 8b, shows that the volume fraction does not change with time, indicating the system has reached the equilibrium state before cyclic loading.

For an intuitive understanding of the relationships between the volume fraction of the void and microscopic failure, Figure 9 represents the time evolution of the spatial distribution of voids for strain rates of (a) 5.0 × 10^8^/s and (b) 5.0 × 10^9^/s. In these figures, some large voids are observed in the late stage of cyclic loading. These voids are considered to be a type of microscopic damage that can be the starting point for macroscopic fractures. At a strain rate of 5.0 × 10^8^/s, noticeably larger voids were generated at approximately 20 cycles, and their volume increased with time. At a strain rate of 5.0 × 10^9^/s, the generation of larger voids was observed at approximately 200 cycles, and subsequent fusions of these voids was observed. The generation and expansion of voids are consistent with the time-dependent behavior of the volume fraction of voids, as shown in Figure 8b.

### 3.5. Residual Strength

Finally, in Figure 10, the stress–strain curves are presented for uniaxial tensile calculations with three different initial structures: the equilibrium structure without cyclic loading, the structure after 100 cycles at a strain rate of 5.0 × 10^8^/s, and the structure after 1000 cycles at a strain rate of 5.0 × 10^9^/s. It should be noted that the structures after cyclic loading are not in equilibrium states. In this figure, the strengths of the two cyclic-loaded structures are approximately 10% lower than that of the structure without cyclic loading. As we have described, voids and disentanglement in the polymer result in microscopic damage and reduced strength. Furthermore, both of the initial structures exhibit almost the same strength after cyclic loading. This may be because these structures have almost the same entropy, as shown in Figure 6.

## 4. Conclusions

The time evolution of the microscopic damage of polymers under cyclic loading is closely related to the durability of the composites. However, a method for quantitatively evaluating microscopic damage has not yet been developed. In this study, full-atomic molecular dynamics simulations were performed to investigate the relationships among the material characteristics associated with microscopic damage: stress–strain curves, entropy, end-to-end distance of the polymer, and volume fraction of the voids. The stress–strain curves show that the polymers behave inelastically with the number of cyclic loadings. The entropy generation, degree of disentanglement of polymers, and total number of voids increase with time, which suggests an increase in microscopic damage, causing inelasticity. Unlike the volume fraction of voids that grow in a nucleation-like manner, the time evolution of entropies does not depend on the strain rate. Two polymer systems with almost the same entropy but different loading conditions exhibit similar reductions in uniaxial tensile strength compared to that of the unloaded polymer system. These results imply that the degradation of the material characteristics is highly dependent on entropy, which is expressed as the sum of all damages.

## Figures and Tables

**Figure 1 polymers-14-04955-f001:**
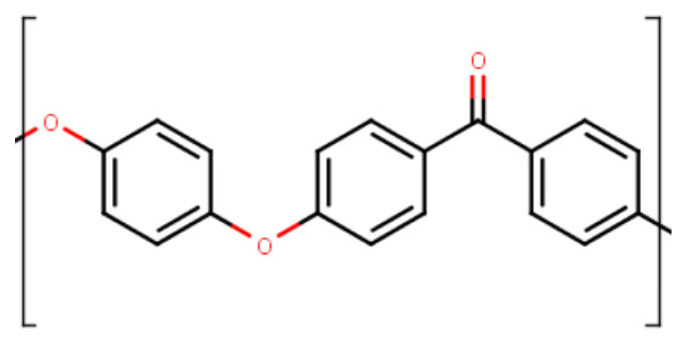
Chemical structure of PEEK.

**Figure 2 polymers-14-04955-f002:**
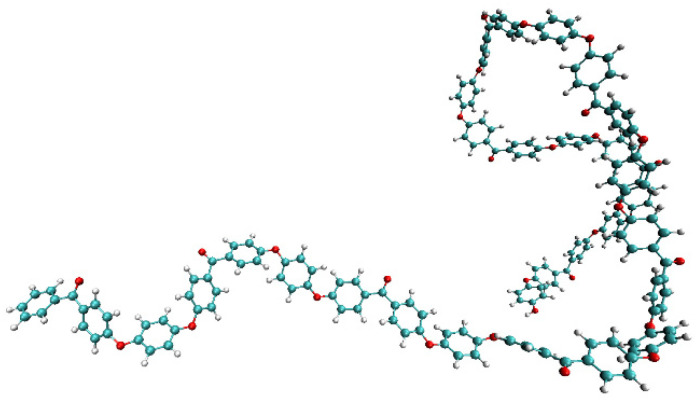
Snapshot of PEEK polymer (*n* = 10).

**Figure 3 polymers-14-04955-f003:**
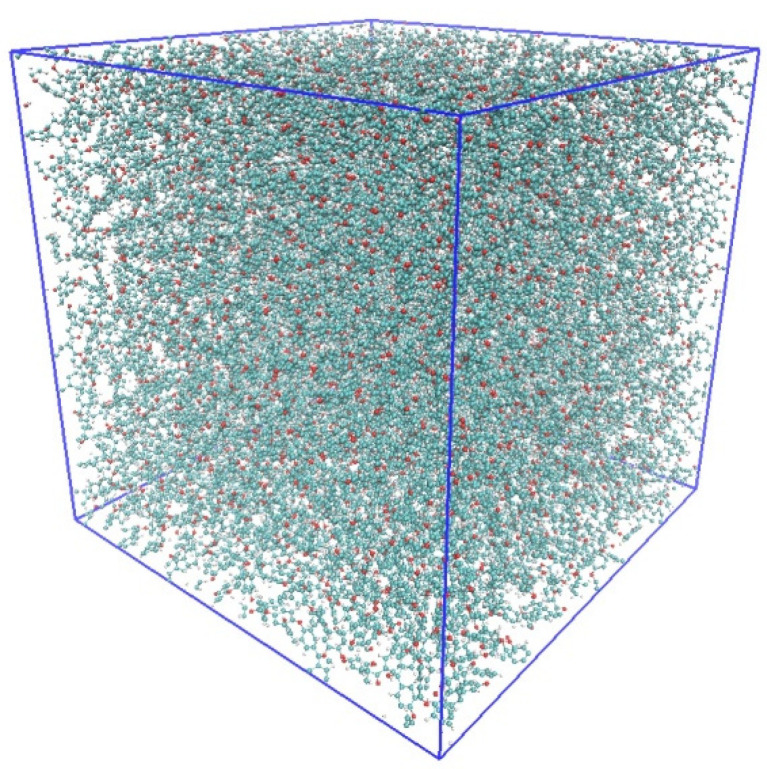
Snapshot of the equilibrium system.

**Figure 4 polymers-14-04955-f004:**
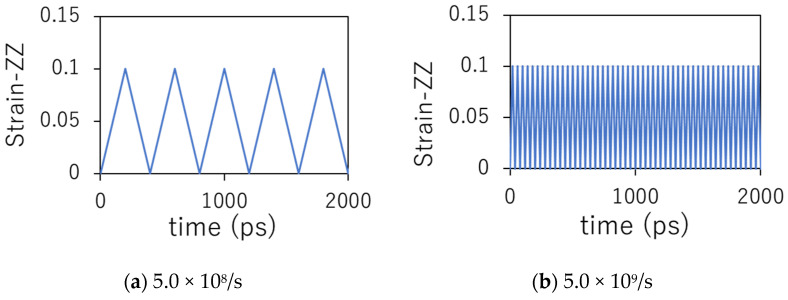
Relationship between time and strain in z-direction during each cyclic loading.

**Figure 5 polymers-14-04955-f005:**
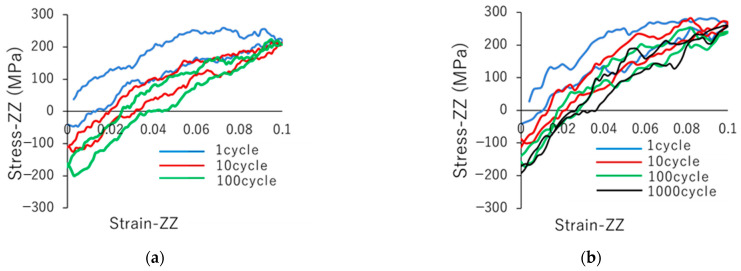
Stress–strain curves for strain rates of (**a**) 5.0 × 10^8^/s and (**b**) 5.0 × 10^9^/s.

**Figure 6 polymers-14-04955-f006:**
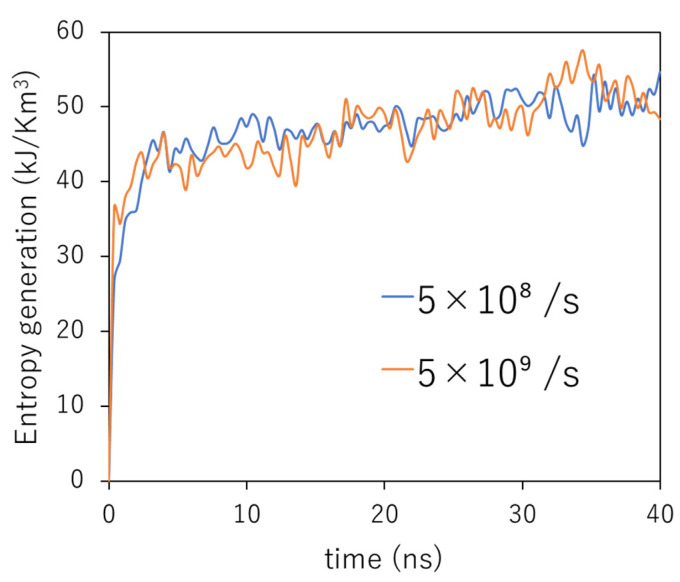
Time development of the entropy generation under a cyclic loading with different strain rates.

**Figure 7 polymers-14-04955-f007:**
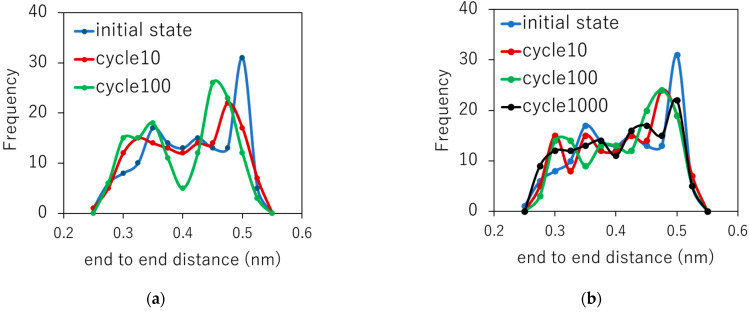
Frequency distribution of R for strain rates of (**a**) 5.0 × 10^8^/s and (**b**) 5.0 × 10^9^/s.

**Figure 8 polymers-14-04955-f008:**
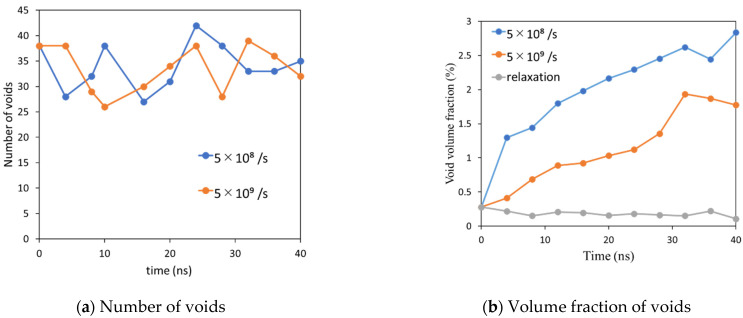
Time evolution of the number and the volume fraction of voids.

**Figure 9 polymers-14-04955-f009:**
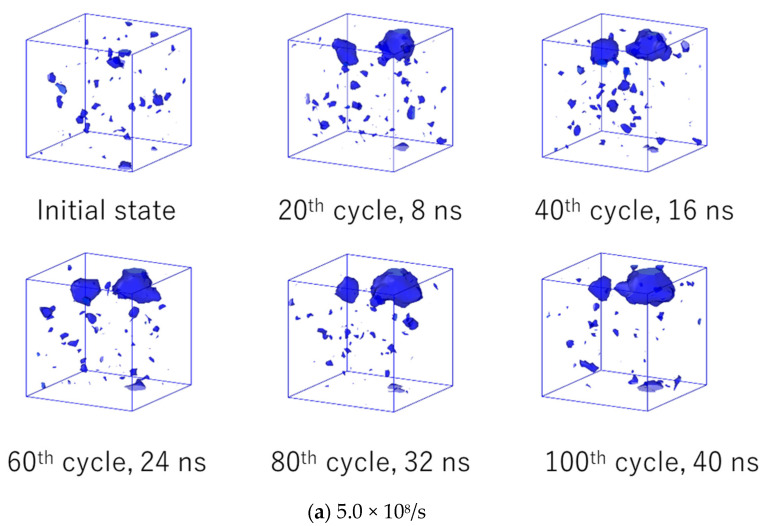
Snapshots of the time evolution of the spatial distribution of voids.

**Figure 10 polymers-14-04955-f010:**
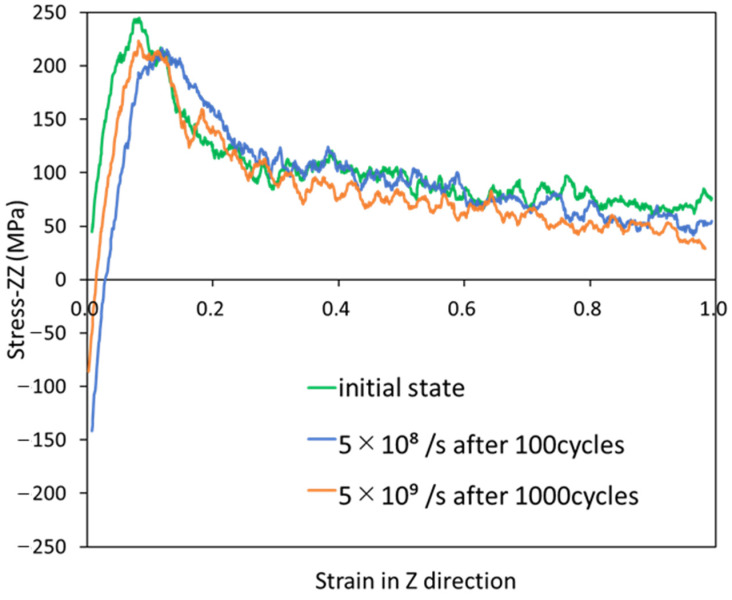
Stress–strain curves for uniaxial tensile calculations with three different initial structures.

## Data Availability

Not applicable.

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
