# Peer review of "Evaluation of Microscopic Damage of PEEK Polymers under Cyclic Loadings Using Molecular Dynamics Simulations"

_polymers, 2022, doi:10.3390/polym14224955_

Round 1

Reviewer 1 Report

In this work, using atomistic simulation Iwamoto, Oya and Koyanagi. study the time evolution of the microscopic damage of polymeric melt under cyclic loading process. The author did a systematic investigation on the materials damage characterization using entropy, chain disentanglements and void expansions during the loading process. The work imply that the material failure is highly related to the entropy gained that summarize all the microscopic changes.

      Overall, this manuscript presents a solid scholar work on an important material failure problem. I personally like the work based on: The motivation/problem is well laid out and of significance (relate the microscopic damage to material failure); The use of entropy gained to quantify the microscopic damage is both ingenious and insightful—It simply a very complicated problem by just using a simple, fundamental  quantity. The results are solid, and the findings are informative. Thus, I support the publication of the article with just few minor points for the authors’ consideration.

1.     The preparations of equilibrium configurations are of significance to this study. I notice that the authors perform four steps of simulation to prepare equilibrated systems. To guarantee the systems reach equilibrium, can the authors show more evidence? For example, does the end-to-end vector time correlation function <R(0) *R(t)> decays to zero within the simulation time?

2.     Thermodynamically, the use of Eqn(1) to calculate entropy is under the assumption that the process is reversible (for example, in infinitely slow process, the systems are always in equilibrium), which is impossible in practical situations. To verify of this approach, I suggest the author verify that the choice of shear rate in these study does not cause major errors for calculating entropy. Maybe a bench-mark test on the entropy-shear rate dependence will help, if the computational time is reasonable.

3.     Section 3.3, the authors use the classic polymer theory to relate the entropy with end-to-end system. I encourage the author explicitly write the relation equation to gives more insight to the readers.

Reviewer 2 Report

The manuscript entitled: ‘’Evaluation of microscopic damage of PEEK polymers under cy-clic loadings using molecular dynamics simulations”, is interesting and scientifically relevant. The described results show that the degree of disentanglement of polymers and the volume fraction of voids increase with cyclic loading, which may lead to entropy generation. Authors claim that the entropy generation is expressed as the total microscopic damage and can potentially be employed for effective evaluation of the degradation of material characteristics.

The manuscript raises an important topic and therefore I recommend the work for publication after minor revision. Below are presented questions and suggestions.

Were there tested the reproducibility of the results on the same type of material and the differences at the results obtained with the use of PEEK with different averge molecular weight?

Figure 5, page 6 - the legend used at the figure a and b are different, it should be corrected. Why in Figure 5a there is a lack of curves for 1000 cycles?
